# Immunotherapy Breakthroughs in the Treatment of Recurrent or Metastatic Head and Neck Squamous Cell Carcinoma

**DOI:** 10.3390/cancers12092691

**Published:** 2020-09-21

**Authors:** Christian Borel, Alain C. Jung, Mickaël Burgy

**Affiliations:** 1Department of Medical Oncology, Institut de Cancérologie Strasbourg Europe, 67200 Strasbourg, France; m.burgy@icans.eu; 2Laboratoire de Biologie Tumorale, Institut de Cancérologie Strasbourg Europe, 67200 Strasbourg, France; a.jung@icans.eu; 3Laboratory STREINTH (Stress Response and Innovative Therapies), Inserm IRFAC UMR_S1113, Université de Strasbourg, 3 av. Molière, 67200 Strasbourg, France

**Keywords:** head and neck squamous cell carcinoma, cancer immunotherapy, immune checkpoint inhibitors, recurrent or metastatic

## Abstract

**Simple Summary:**

Head and neck squamous cell carcinoma (HNSCC), in the locally advanced setting, relapses in more than 50% of cases after surgery and/or chemo-radiotherapy. Prognosis is then very poor with platinum-based chemotherapy yielding a median survival of 10 months and a 2-year survival rate of less than 20%. In recent years, immunotherapy, has changed treatment and prognosis of several tumor types. Given their inflammatory profile and high mutational burden, HNSCC is a good candidate for immunotherapy. This review describes the major therapeutic breakthrough worked by Immune Checkpoint Inhibitors (ICIs) blocking the PD-1/PD-L1 axis. These monoclonal antibodies have doubled the 2-year survival rate thanks to long lasting responses with a favorable toxicity profile. Further progress is expected with new immunotherapeutic agents having a different mechanism of action combined with anti-PD(L)1, chemotherapy or targeted therapies.

**Abstract:**

Head and neck squamous cell carcinoma (HNSCC) in the recurrent or metastatic (R/M) setting is a devastating disease with a poor prognosis. Until recently, the reference first line treatment was the EXTREME protocol, which yields a 10.1 months median survival, and almost no effective treatment are available in second line. Immune checkpoint inhibitors (ICIs) have changed the prognosis of several metastatic solid tumors. Given their inflammatory profile and high mutational burden, HNSCC is a good candidate for ICIs treatments. First, a strong pembrolizumab efficacy signal was shown in the Keynote-012 Phase Ib study. Then, the phase III Checkmate-141 study validated the efficacy of nivolumab in platinum-resistant patients. Finally, the first line conquest is acquired since the final results of the keynote-048 phase III study that demonstrated the superiority of pembrolizumab versus EXTREME in CPS ≥ 1 patients, and with the addition of platinum and 5FU in all patients. However, the first line treatment landscape is not frozen. Two studies (Checkmate-651 and Kestrel) are investigating the efficacy of the combination of antibodies raised against CTLA-4 and PD-(L)1. Results are impatiently awaited. Further progress needs the use of new immunotherapeutic agents such as monalizumab or ICOS agonist rather in combination with an anti-PD(L)1. New associations of ICIs and chemotherapeutic or targeted therapeutic agents are also actively investigated. Finally, ICIs has to be studied in the locally advanced setting where there is a chance of cure. Several trials are testing the potential synergistic combination of ICIs with radiotherapy and platinum or cetuximab, or ICIs used in a neoadjuvant setting.

## 1. Introduction

Head and neck squamous cell carcinoma (HNSCC) in the recurrent or metastatic (R/M) setting has a poor prognosis. Until recently the EXTREME regimen had been the standard of care for patients considered as platinum-sensitive with a median survival of 10.1 months [1]. For patients pretreated with platinum, second line options are few (typically: methotrexate, cetuximab or taxanes) with a response rate that varies between 3% and 13% and a median survival inferior to 6 months [2,3,4]. HNSCC are frequently characterized by an inflammatory tumor profile with lymphocytic infiltration and a strong PD-L1 expression on tumor cells and on tumor micro-environment (TME) cells alike. In such tumors, lymphocytes, particularly T helpers, induce the secretion of gamma interferon which stimulates the expression of PD-L1 on the TME cells thus protecting tumor cells from the action of cytotoxic T cells. Preclinical studies have thus shown that blocking the interaction between PD-1 and its ligand PD-L1 increases the activation of cytotoxic T cells and inhibits tumor growth [5].

## 2. KEYNOTE-012 Phase 1b Study: Anti-Tumor Activity of Pembrolizumab

Pembrolizumab is a highly selective humanized monoclonal immunoglobulin G4 which blocks the interaction between PD-1 and its ligand PD-L1. The anti-tumor activity has been at first demonstrated in a cohort of 60 HNSCC R/M PD-L1 positive patients with a dose of 10 mg/kg every 2 weeks. An overall response rate (ORR) of 18% was observed. Above all, most of these responses were durable with a median of 53 weeks, a median PFS of 2 months (95% CI: 2–4) and a particularly interesting median survival of 13 months (95% CI: 5 – not reached) [6].

The expansion cohort was later conducted with 132 patients whatever was their PD-L1 and HPV status, with a reduced dose and longer intervals between administrations: total dose of 200 mg every 3 weeks which will become the standard schedule of administration for pembrolizumab. The expansion cohort was a heavily pretreated population since 82% of patients were at least in second line and 57% were in third line. An ORR of 18% (95% CI: 12–26) was confirmed (central review). As in the initial cohort, responses are durable but follow-up was insufficient to allow a precise evaluation of the duration to be made (median not reached (≥2 months, ≥11 months), an excellent tolerance was observed since treatment related adverse events of grade 3–4 were only 9%. The determination of the expression of PD-L1, not just on tumor cells but also on TME cells is particularly demonstrative since the combined determination is predictive of response (ORR: 22% if PD-L1 ≥ 1% vs. 4% if PD-L1 < 1% *p* = 0.021) and of survival (OS:10 months if PD-L1 ≥ 1% vs. 5 months if PD-L1 < 1%) [7].

Pooled results of the initial cohort and of the expansion cohort were published with long term follow-up (median 9 months (range, 0.2–32)) [8]. Treatment related adverse events of any grade and grade 3–4 occurred in 64% and 13% of patients respectively. ORR is 18% (95% CI, 13–24) and does not depend much on prior treatment: 17% after platinum, 15% after cetuximab. Duration Of Response (DOR) is high with a not reached median (range, 2+ to 30+ months), 71% of responses lasted more than 12 months and five patients completed the study after 2 years of treatment with pembrolizumab. Median survival is 8 months (95% CI, 6–10 months) with a 1 year survival rate of 38%, which compares favorably with a second line treatment with cetuximab (1-y OS: 11%) [3] or with methotrexate (1-y OS: 28%) [2]. Thus, it is probable that pembrolizumab does not benefit responsive patients only. The PD-L1 expression according to the combined positive score (CPS): tumor and TME cells, is predictive of the response rate (21% if CPS ≥ 1 vs. 6% if CPS < 1 *p* = 0.023) and of survival (median 10 months if CPS ≥ 1 vs. 5 months if CPS <1), whereas PD-L1 expression according to the tumor proportion score (TPS) which takes into account tumor cells only is not. ORR and survival are independent of the HPV status.

Because of an excellent tolerance, of a response rate comparing favorably with what is usually achieved by standard second line therapies and above all of a very long duration of response, FDA granted accelerated approval to pembrolizumab for patients with R/M HNSCC.

## 3. Phase II and III Second Line Studies and Beyond

### 3.1. Phase II Studies

Phase II studies did confirm the results of phase I for pembrolizumab as for durvalumab (Table 1**)**. The KEYNOTE-055 study is a very wide phase II study evaluating the response rate and toxicity of pembrolizumab at 200 mg every 3 weeks: 171 patients resistant to platinum and cetuximab were included. Response rate was 16% (95% CI: 11–23), grade 3–4 adverse events 15%, which confirmed results of KEYNOTE-012. As in KEYNOTE-012 responses are durable with an 8 months median (2+–12+) and 75% of responses still ongoing at the time of analysis [9].

Concerning durvalumab (selective high-affinity engineered IgG1 mAB which blocks PD-L1 to PD-1), its efficacy was confirmed by the Hawk and Condor studies. HAWK included 112 strict second line platinum resistant patients with PD-L1 high expression (TC ≥ 25% using Ventana PD-L1 (SP263) assay) [10]. Durvalumab exhibited a significant anti-tumor activity, ORR coming out at 16.2% (95% CI: 9.9–24.4). To be noted that contrariwise to what has been observed with pembrolizumab, ORR is significantly better in HPV+ patients (29.4%) than in HPV– patients (10.9%), which translates into a survival advantage. Toxicity is particularly low at 8% of Grade ≥ 3 AE. Median PFS was 2.1 months (95% CI: 1.9–3.7) and Median OS 7.1 months (95% CI: 4.9–9.9) similar to that observed with other anti-PD-(L)1 agents.

CONDOR is a phase II randomized study for patients with low/negative PD-L1 expression (TC < 25%) who have progressed after 1 platinum containing regimen in the R/M setting [11]. This study comprises 3 arms with a randomization 2:1:1: durvalumab + tremelimumab (D + T), durvalumab (D) and tremelimumab (T: anti-CTLA-4). Primary endpoint is ORR and tolerance particularly for the D + T association in order to determine whether the dual blockade of PD-L1 and cytotoxic T-Lymphocyte associated protein 4 (CTLA-4) may overcome immune checkpoint inhibition. 267 patients have been randomized. Tolerance is acceptable, including in the arms which features the anti-CTLA-4: the rate of Grade 3–4 AEs being 15.8%, 12.3% and 16.9% respectively. Contrariwise, the addition of T does not allow an increase of the ORR: 7.8% with D + T, 9.2% with D and 1.6% with T, respectively. This weak rate of response may be explained by the low expression of PD-L1, however a clinical benefit seems to exist in terms of survival with results comparing with what is usually reported in second line treatments with anti-PD-1: median OS 7.6 months and 6 months, 1-y OS 37% and 36% for D + T and D, respectively.

### 3.2. Phase III Studies

Phase III studies have validated a survival benefit with nivolumab and pembrolizumab vs. a standard monotherapy for diseases progressing less than 6 months after a platinum-based CT (Table 2)**.**

#### 3.2.1. Nivolumab and the CHECKMATE-141 Study

Nivolumab is a fully human IgG4 anti-PD-1 monoclonal antibody that has shown anti-tumor efficacy in multiple tumor type especially in advanced squamous-cell non-small-cell lung cancer [12]. Study CHECKMATE-141 includes 361 patients progressing less than 6 months after a platinum-based CT and randomizes with a 2:1 ratio a treatment with Nivolumab (3 mg/kg q2w) vs. a standard monotherapy (methotrexate, weekly docetaxel, cetuximab) [13]. Primary endpoint being overall survival is met with a 7.5 months median (95% CI: 5.5–9.1) in the nivolumab arm vs. 5.1 months (95% CI: 4.0–6.0) in the standard arm. The difference is significant (HR for death 0.70, 97.73%CI, 0.51–0.96; *p* = 0.01), 1 year survival rate is 36% in the nivolumab arm vs. 16.6% in the standard arm with a favorable toxicity profile: 13.1% of treatment related AEs in the nivolumab arm vs. 35.1% in the standard arm. There is however a dissociation of results between PFS and OS since PFS does not differ significantly from one arm to the other (median PFS 2.0 months for nivolumab vs. 2.3 months for the standard therapy HR 0.89; 95% CI, 070 to 1.13; *p* = 0.32). Response rate is also moderate (13.3% for nivolumab vs. 5.8% for standard therapy). 

**Table 1 cancers-12-02691-t001:** Phase Ib-II in R/M pretreated patients (platinum resistant).

Studies	IO Agent	Phase	Nb of Patients	ORR (95%CI)	DOR Median Mo (Extremes)	Grade ≥ 3 Related Toxicities	PFS Median Mo (95%CI)	OS Median Mo (95%CI)	1-Year OS	Remarks
KEYNOTE-012 [6]	P	Ib Initial cohort	60	18%	53 weeks	17%	2 (1–4)	13 (5-NR)		PD-L1 positive
KEYNOTE-012 [7]	P	Ib expansion cohort	132	18% (12–26)	NR (2–11)	9%	2 (2–2.2)	8 (6–10)		Wathever PD-L1, HPV stat.
KEYNOTE-012 [8]	P	Ib: pooled	192	18% (13–24)	NR (2+–30+)	13%		8 (6–10)	38%	Long term FUP
Study 1108 [14]	D	Ib-II	62	11% (4–21)	6 resp. ≥12 mo	8%		8.9 (5.4–18.8)	42%	79% of pt ≥3rd line
KEYNOTE-055 [9]	P	II	171	16% (11–23)	8 (2+–12+)	15%	2.1 (2.1–2.1)	8 (6–11)		Platinum Cetuximab Refractory
HAWK [10]	D	II	112	16.2% (9.9–24.4)	10.3 mo	8%	2.1 (1.9–3.7)	7.1 (4.9–9.9)	33.6%	PD-L1 high
CONDOR [11]	D T	II randomized	267	D + T: 7.8% D: 9.2% T: 1.6%	9.4(4.9-NR)	15.8% 12.3% 16.9%	2.0 1.9 1.9	7.6(4.9–10.6 6(4–11.3) 5.5(3.9–7)	37% 36% 24%	PD-L1 Low/neg.

P: pembrolizumab; D: durvalumab; T: tremelimumab; NR: not reached; FUP: follow-up; IO: immuno oncology; ORR: Overall Response Rate; DOR: duration of response; mo: months; PFS: Progression Free Survival; OS: Overall Survival.

**Table 2 cancers-12-02691-t002:** Phase III studies in R/M platinum resistant patients.

Studies	Number of Patients	OS: Median Mo (95%CI)	1-Year OS (95% CI)	2-Year OS (95%CI)	PFS Median Mo (95% CI)	ORR (95% CI)	Related Grade ≥ 3 Toxicities
CHECKMATE-141 [13,15]	N: 240 IC:121	N: 7.5 (5.5–9.1) IC: 5.1 (4.0–6.0) HR, 0.70 (0.51–0.96) * *p* = 0.01	N: 36% (28.5–43.4) IC: 16.6% (8.6–26.8)	N: 16.9% (12.4–22) IC: 6.0% (2.7–11.3)	N: 2.0 (1.9–2.1) IC: 2.3 (1.9–3.1) HR, 0.89 (0.7–1.13) *p* = 0.32	N: 13.3% (9.3–18.3) mDOR: 9.7 mo IC: 5.8% (2.4–11.6) mDOR: 4.0 mo	N: 13.1% IC: 35.1%
KEYNOTE-040 [16]	P: 247 IC: 248	P: 8.4 (6.4–9.4) IC: 6.9 (5.9–8.0) HR, 0.80 (0.65–0.98) *p* = 0.0161	P: 37.0% (31.0–43.1) IC: 26.5%(21.2–32.2)		P: 2.1 (2.1–2.3) IC: 2.3 (2.1–2.8)	P: 14.6% (10.4–19.6) mDOR: 18.4 mo IC: 10.6% (6.6–14.5) mDOR: 5.0 mo *p* = 0.061	P: 13% IC: 36%
EAGLE [17]	D: 240 D + T: 247 SoC: 249	D: 7.6 (6.1–9.8) D + T: 6.5 (5.5–8.2) SoC: 8.3 (7.3–9.2) D vs. SoC: HR, 0.88 (0.72–1.08) *p* = 0.20 D + T vs. SoC: HR 1.04 (0.85–1.26) *p* = 0.76	D: 37% (30.9–43.1) D + T: 30.4% (24.7–36.3) SoC: 30.5% (24.7–36.4)	D: 18.4% (13.3–24.1) D + T: 13.3% (8.9–18.6) SoC: 10.3% (5.7–16.5)	D: 2.1 (1.9–3.0) D + T: 2.0 (1.9–2.3) SOC: 3.7 (3.1–3.7)	D: 17.9% (13.3–23.4), mDOR: 12.9 mo D + T: 18.2% (13.6–23.6), mDOR: 7.4 mo SoC: 17.3% (12.8–22.5), mDOR: 3.7 mo	D: 10.1% D + T: 16.3% SoC: 24.2%

OS: overall survival; mo: month; CI: confidence interval; PFS: Progression Free Survival; ORR: Overall Response Rate; N: nivolumab; IC: Investigator Choice; HR: Hazard Ratio; *: 97.73% CI; mDOR: median duration of response; P: pembrolizumab; D: durvalumab; T: tremelimumab; SoC: Standard of Care.

Weak PFS and moderate response rate contrast with the 1 year survival rate which doubles in the Nivolumab arm, which suggests that responding patients are undoubtedly not alone in benefiting from the treatment with Nivolumab and that the RECIST 1.1 criteria are not fully adapted to the evaluation of the efficacy of immunotherapy, which may result in premature discontinuations of treatment. The CHECKMATE-141 study, like many immunotherapy studies, authorized continuation of treatment beyond progression defined by the RECIST 1.1 criteria, under the condition of a stable ECOG PS, of a clinical benefit and of the absence of rapid progression. Post-hoc analysis of patients treated beyond progression has been reported [18]: 62 patients out of the 146 showing a progression were treated beyond progression with nivolumab: 25% achieved a stability, 25% a reduction of target lesions < 30% and 5% a reduction of target lesions > 30%.

Long term results of the CHECKMATE-141 (minimum follow-up 24.2 months) confirm a survival advantage (HR 0.68; 95% CI: 0.54–0.86) with a 2 years survival rate almost tripled: 16.9% in the nivolumab arm vs. 6.0% in the standard arm [15]. The survival benefit seems to be more important for PD-L1 ≥ 1% patients (HR 0.55; 95% CI: 0.39–0.68) but may be observed also for PD-L1 < 1% patients (HR 0.73; 95% CI: 0.49–1.09). Survival benefit is independent from HPV status.

The results of the nivolumab arm of the CHECKMATE-141 study have been confirmed in a real-world population i.e., the TOPNIVO study, the interim analysis of which have been reported at ASCO 2019 on 199 patients: 19% of patients were older than 70 and 16% were PS 2. The overall survival benefit is similar with a 7.7 median OS (95% CI: 6.0–9.5) and no additional toxicities were observed with 10.5% grade ≥ 3 related AEs. Moreover, this study provides additional survival data in subgroups of patients, more particularly in older patients who benefit from the same improvement in survival (median OS 7.0 months (95% CI: 4.7-NR) [19].

#### 3.2.2. Pembrolizumab and KEYNOTE-040 Study

Based on the solid results of KEYNOTE-012 and of KEYNOTE-055, KEYNOTE-040 is their logical next step [16]. Very similar in its design to CHECKMATE-141 randomizing pembrolizumab at 200 mg every 3 weeks, but with a 1:1 ratio, vs. a standard treatment left to the choice of the investigator between methotrexate, docetaxel or cetuximab. Population of KEYNOTE-040 may be a little more favorable than in CHECKMATE-141 as it includes patients having progressed during or after a treatment with platinum but without any time limit and with a maximum of 2 lines of treatment in the R/M setting or within 3 to 6 months after a multimodal treatment in the LA setting, which eliminates very rapid progressions within 3 months. 14% to 16% of enrolled patients showed disease progression within 3–6 months of a previous cisplatin based multimodal therapy for locally advanced disease, 57% had received a first line platinum based chemotherapy in the R/M setting some of whom could be considered as platinum sensitive if they had progressed more than six months after the last dose of platinum, and finally 27% to 28% are in third line in the R/M setting. Primary endpoint being overall survival in the pembrolizumab arm is met with a median of 8.4 months vs. 6.9 months in the standard arm and with a HR of 0.80 (95% CI: 0.65–0.98) *p* = 0.0161, 1-year survival is 37% in the pembrolizumab arm vs. 26.5% in the standard arm.

If the HR seems to be less favorable than in CHECKMATE-141, this is not due to an inferior performance of the pembrolizumab arm but rather to an over-performance of the standard arm when compared to CHECKMATE-141 (Table 2). 45% of patients in the standard arm had been treated with docetaxel which is more efficient than methotrexate or cetuximab vs. 43% in CHECKMATE-141, which is roughly equivalent, however docetaxel is administered q3w in KEYNOTE-040 which is a more efficient mode than the weekly administration in CHECKMATE-141. Above all, 13% of patients in the standard arm of KEYNOTE-040 had benefited from a third or fourth line of immunotherapy which increases survival furthermore.

Last but not least, the great added interest of KN-040 is to have confirmed a biomarker which is predictive of efficacy and survival: the combined positive score (CPS) of the PD-L1 expression, whereas the tumor proportion score (TPS) with a 1% cut-off of CHECKMATE-141 is not. Indeed, for CPS ≥ 1 median OS is 8.7 months in the pembrolizumab arm vs. 7.1 months in the standard arm (HR 0.74 (95% CI: 0.58–0.93) *p* = 0.0049) whereas for CPS < 1 median OS is 6.3 months in the pembrolizumab arm vs. 7.0 months in the standard arm (HR: 1.20 (95% CI: 0.8–2.07) *p* = 0.84). The expression of PD-L1 measured by TPS does not become predictive until high values ≥ 50% are reached with a median survival of 11.6 months in the pembrolizumab arm vs. 6.6 months in the standard arm (HR 0.53 (95% CI: 0.35–0.81) *p* = 0.0014), whereas for TPS < 50% median survival is of 6.3 months in the pembrolizumab arm vs. 7.0 months in the standard arm (HR: 0.93 (95% CI: 0.73–1.17) *p* = 0.26).

Response rate remains moderate at 14.6% in the pembrolizumab arm vs. 10.1% in the SOC arm, but with a very long median response duration in the pembrolizumab arm of 18.4 months. Toxicity profile is also in favor of the pembrolizumab arm with a rate of treatment related Grade ≥ 3 AE of 13% vs. 36% in the standard arm.

#### 3.2.3. Durvalumab with or without Tremelimumab vs. Standard of Care: the EAGLE Study 

Chronologically EAGLE [17] comes out after CHECKMATE-141 and KEYNOTE-040. Supported by solid phase Ib data [14], its purpose is not limited to measuring the efficacy of durvalumab (D) alone, in second line after progression after one, and only one, cisplatin based first line, but also in conjunction with an anti-CTL-4, tremelimumab (T) vs. standard of care (SoC: cetuximab, weekly paclitaxel or docetaxel, methotrexate or a fluoropyrimidine). Primary endpoint is OS: D vs. SoC and D + T vs. SoC. 736 patients have been randomized: 240 in the D arm, 247 in the D + T arm and 249 in the SoC arm. It is a negative study without any significant increase in survival, be it in the D arm HR 0.88 (95% CI: 0.72–1.08; *p* = 0.20) or in the D + T arm HR 1.04 (95% CI: 0.85–1.26; *p* = 0.76). Observed median, 1-year and 2-year survivals with durvalumab are however comparable to what was obtained with Nivolumab in study CHECKMATE-141 and with pembrolizumab in study KEYNOTE-040, i.e., 7.6 months, 37% and 18.4% (Table 2). Response rates of 17.9% for D and 18.2% for D + T evidence a clinical activity for durvalumab whereas the addition of T does not improve the results: such responses as is usual with immunotherapy are long-lasting (median DOR 12.9 months (6.9–21) for durvalumab vs. 3.7 (2.0–4.2) for SoC). In the end, the negativity of the study is attributable to an over-performance of the SoC arm when compared to what is usually observed. This over-performance in the SoC arm is attributable to the fact that patients are on strict second line, to an excess of patient PS 0 or suffering solely from a metastatic disease, to a large number of patients who have been able to receive weekly paclitaxel (the most efficient mode of treatment of the SoC) and last who have been able to receive an immunotherapy at a later stage. To be noted also over-mortality in the initial part of the immunotherapy arms survival curves, which raises the question of predictive efficacy markers aiming at making a better selection of patients most suitable for immunotherapy. The evaluation of the PD-L1 expression on tumor cells (TC) is not predictive of survival. Contrariwise, in a post-hoc analysis, peripheral blood tumor cell mutation burden (bTMB) is predictive of survival: for patients having a mutation burden ≥16 mut/MB median survival increases from 4 months with the standard chemotherapy to 7.6 months with D + T (HR 0.38(95% CI: 0.19–0.78) *p* = 0.0061) and to 8.1 months with D (HR 0.39 (95% CI: 0.2–0.76) *p* = 0.0044) [20]. Thus bTMB confirms the findings of a retrospective study in 126 HNSCC patients where the response rate and overall survival were correlated with TMB values (TMB: 21.3 mut/MB in any responders vs. 8.2 mut/MB in non-responders *p* = 0.01 and median survival: 20 months if TMB > 10 mut/MB vs. 6 months if TMB < 5 mut/MB *p* = 0.01) [21]. In addition, TMB and inflammatory biomarkers are independent predictors of response and survival with pembrolizumab [22]. However the TMB cut-off has to be more precisely defined and a prospective trial has to be conducted in order to validate his predictive value.

Last, toxicity profile is in favor of durvalumab: 10.1% of treatment related Grade ≥ 3 AE for the durvalumab arm vs. 16.3% for the D + T arm and 24.2% for the SoC arm.

### 3.3. Conclusions for Platinum-Resistant Patients

Phase II and III studies have confirmed the clinical efficacy of anti-PD-(L)1 which had been suggested by phase I studies, particularly KEYNOTE-012. Phase III studies have demonstrated the superiority for patient previously treated with platinum at first of Nivolumab (CHECKMATE-141 study) and later of pembrolizumab (KEYNOTE-040 study) versus SoC, which resulted in an approval by the FDA at first and by the EMA later restricted to patients TPS ≥ 50% for pembrolizumab. Although response rates remain moderate < 20%, the two highlights are: the duration of response to immunotherapy > 12 months and the 2-year survival which is more than doubled. The positivity of the studies does not depend on the anti-PD-(L)1, the efficacies of which are equivalent in terms of response and survival rates, but of the performance of the SoC arm, which explains the less favorable HR of KN-040 and the negativity of EAGLE which besides does not validate an increase of efficacy through the combination of an anti-PD-L1 with an anti-CTLA-4. Indeed, an anti-CTLA-4 combined to anti-PD-L1 does not seem to be synergistic in spite of a solid scientific rational. Lymph nodes surgical resection or irradiation might impair the anti-CTLA-4 action. Furthermore, tremelimumab being an IgG2 is not able to induce anti-body-dependent cell toxicity whereas NK cell are the most represented T-cells in the TME of HNSCC [23].

The logical next step is to study the efficacy of anti-PD-(L)1 for platinum sensitive patients, i.e., in first line R/M with two imperatives: increase the efficacy and research biomarkers predictive of response, the expression of PD-L1 as measured by the TPS not bring discriminatory.

## 4. First Line Studies: Platinum Sensitive Patients

First line studies are trying to improve the efficacy of anti-PD-L1 monotherapies demonstrated for platinum resistant patients in two ways: either by combining immunotherapy to standard chemotherapy (KEYNOTE-048 study), or by combining 2 antibodies targeting two different pathways of immune escape, specifically in associating an anti-PD-(L)1 with an anti-CTLA-4 (CHECKMATE-651 and KESTREL study). Actually the addition of a standard chemotherapy may induce an immunogenic cell death, increases the release of neo-antigens and increases lymphocytic infiltration of the tumor micro-environment (TME) [24]. Moreover, chemotherapy disrupts tumor architecture which reduces immune exclusions areas and allows a quick tumor control to be achieved [25]. It is a strategy which has already demonstrated its efficacy in other cancer types [26,27,28,29].

### 4.1. Pembrolizumab (P) or Pembrolizumab + Chemotherapy (C: platinum + 5Fu) vs. EXTREME: KEYNOTE-048 Study

It is a 3 arms phase III study, comparing P (301 patients) vs. EXTREME (300 patients) on the one hand and P + C (281 patients) vs. EXTREME (278 patients) on the other hand, for patients in the R/M setting considered as platinum-sensitive, i.e., patients having received their last administration more than 6 months before, within the course of the multi-modal treatment of the primary tumor, with overall survival as a primary endpoint.

Preliminary results were reported at ESMO 2018 [30], final results at ASCO 2019 [31] and were finally published in *The Lancet* [32]. Complementary data have been communicated in the EMA report concerning in particular some patient sub-groups with a CPS < 1 and with a CPS between 1 and 19 [33]. Actually results had been stratified initially on the basis of TPS, but in the end survival data and response rates were analyzed on the basis of CPS. Statistically, two comparisons have been scheduled: P vs. EXTREME (Table 3) on the one hand and P + C vs. EXTREME (Table 4) on the other hand.

#### 4.1.1. Pembrolizumab vs. EXTREME

Over the entire population in intent to treat (ITT), pembrolizumab is not inferior to EXTREME with a median survival of 11.5 months vs. 10.7 months. However for CPS ≥ 1 and CPS ≥ 20 patients, overall survival is significantly increased in the pembrolizumab arm with medians of 12.3 and 14.8 months respectively vs. 10.3 and 10.7 months respectively in the EXTREME arm. 2-year survival reaches 35.3% for CPS ≥ 20 patients in the pembrolizumab arm vs. 19.1% in the EXTREME arm.

Response rate to pembrolizumab remains moderate at 16.9% (comparable to the ORR of platinum resistant patients) vs. 36% for the EXTREME arm, however responses achieved in the pembrolizumab arm are very durable with a median duration of 22.6 months whereas median duration of chemotherapy induced responses comes out at 4.5 months.

Contrariwise, the rate of progressive disease is high at 40.5% with pembrolizumab vs. 12.3% in the EXTREME arm which explains the peculiar profile of the PFS curves and impacts the survival profile (excess of early deaths in the pembrolizumab arm): the pembrolizumab survival curve is below the chemotherapy curve over the first eight months, at which point it crosses it to stay constantly above afterwards. This high rate of progressive disease (40.5%) could be partly explained by the phenomenon of hyperprogression where the disease seems to grow faster after initiation of immunotherapy. Hyperprogression has been described in several tumor types. Rates of hyperprogression range from 4% to 29% depending on the definition and the methods used to assess tumor growth kinetics [34]. In HNSCC it may be as high as 29.5% [35], with the consequence of having to change treatment quickly in the face of rapid tumour progression, especially in case of clinical deterioration. However, the concept of hyperprogression is controversial, given that it is not possible to determine whether the acceleration of growth kinetics is due to immunotherapy or reflect the natural history of cancer. Whatever the reason, it may be risky to administer pembrolizumab alone to a patient presenting heavy symptoms and in need of an urgent therapeutic response: for such patients the association of pembrolizumab with chemotherapy is undoubtedly better suited, as will be seen hereunder. Conversely, for patients presenting light symptoms and slow progression, pembrolizumab is an excellent option with a favorable toxicity profile, under the condition that CPS be ≥1. There again, as for platinum-resistant patients, there is a dissociation between PFS which is not significantly different and the significant survival benefit.

Survival data concerning patient sub-groups CPS < 1 and CPS in the 1–19 range have neither been reported to ESMO 2018 nor to ASCO 2019 because not planned in the statistical plan. These data are nevertheless important because one may wonder whether the favorable results of CPS ≥ 1 patients are not attributable in a large proportion to CPS ≥ 20 patients. Results for these sub-groups are published in the EMA report [33]: there is a clinical activity of pembrolizumab for patients in the CPS range 1–19 with an ORR of 14.5%, a median survival of 10.8 months and a survival curve practically identical to the curve of EXTREME, factors in favor of a treatment by pembrolizumab being then a favorable toxicity profile and long-lasting responses. Conversely, the activity of pembrolizumab for CPS < 1 patients seems to be very weak with a response rate of 4.5% and a median survival of 7.9 months. In the end, CPS seems to be a response (ORR: 4.5%, 14.5% and 23.3% with CPS < 1, 1–19, and ≥20 respectively) and survival (median OS: 7.9, 10.8 and 14.8 months with CPS < 1, 1–19, and ≥20 respectively) predictive marker (Table 3).

The impact of the second line is reflected by the PFS2 which is the cumulated progression-free survival of the 2 first lines. Whereas PFS is not significantly different between the pembrolizumab arm and the standard arm, PFS2 differ with a median of 9.4 months (P) vs. 8.9 months (SOC) (HR 0.81 (95% CI: 0.67–098) *p* = 0.017) which justifies the strategy of resorting to pembrolizumab as early as first line [36].

#### 4.1.2. Pembrolizumab (P) + Platinum/5Fu (C) vs. EXTREME

The combination of pembrolizumab with platinum + 5FU induces a significant improvement of overall survival with a median of 13.0 months vs. 10.7 months in the standard arm over the entire population in ITT. The great advantage of adding chemotherapy to immunotherapy is an increase of the response rate to 35.6% and a decrease of the rate of progressive disease to 17.1 %, which is particularly important for patients in need of an urgent therapeutic response. It follows that the excess of early deaths in the initial part of the survival curve with pembrolizumab alone is eliminated in the P + C arm and that after 8 months the P + C curve gains markedly over the EXTREME curve. This comes at the price of an increased toxicity which becomes equivalent to toxicity in the EXTREME arm (85% grade ≥ 3 AEs in P + C vs. 83% in EXTREME). 

However, the P + C combination does not seem to be synergistic as evidenced by equivalent response rates (36.3% in the EXTREME arm) and a sharply decreased duration of response (median of 6.7 months) when compared to pembrolizumab alone (median 22.6 months). For combination P + C, CPS is also a survival predictive factor: for CPS < 1 there is not any survival advantage in resorting to P + C vs. EXTREME (median 11.3 months vs. 10.7 months HR.21), whereas there is a significant advantage for the CPS 1–19 (median 12.7 months vs. 9.9 months HR 0.71 *p* = 0.00726), which is even better for the CPS ≥ 20 (14.7 months vs. 11.0 months HR 0.60 *p* = 0.0004).

Whereas PFS is not significantly different over the entire population, PFS2 is in favor of P + C (10.4 months vs. 9.0 months HR 0.74 *p* = 0.00081) which seems to justify resorting to immunotherapy as early as first line, in combination with chemotherapy if need be [36]. Improvement of PFS2 seems to demonstrate that a second line treatment particularly with immunotherapy following a first line chemotherapy may not be as efficient as an immunotherapy combined with a chemotherapy as early as first line. However, KEYNOTE-048 is not a therapeutic sequences study, less of 50% of patients have received a second line and 16% only received an immunotherapy after chemotherapy.

#### 4.1.3. KEYNOTE-048 Study Conclusion

KEYNOTE-048 is the first immunotherapy in first line study in the treatment of R/M HNSCC to publish its results: this study is positive in overall survival for CPS ≥ 1 patient with pembrolizumab alone and for the entire population with pembrolizumab + chemotherapy vs. EXTREME thus emerging as the new standard of care. These results have led to the issue of an approval by the FDA for pembrolizumab for CPS ≥ 1 patients and for pembrolizumab + chemotherapy for the entire population. These results have also led to the issue of an approval by the EMA but restricted to CPS ≥ 1 patients, be it for pembrolizumab alone or for pembrolizumab + chemotherapy. Most noteworthy is the achievement of durable responses with a 2-year survival rate of 29% to 35% whereas it is less than 20% with the EXTREME regimen, moreover with a favorable toxicity profile when pembrolizumab is administered alone. With immunotherapy, the survival curve starts to level off at the 2-year point which justifies an attempt to stop treatment, a rather unique result for R/M HNSCC. There remains however quite a lot to be improved on since response rate does not exceed 23% and since progressive disease rate is 40.5% with pembrolizumab alone: it is imperative to improve efficacy and to make a better selection of patients able to best benefit from immunotherapy. The association of immunotherapy with chemotherapy is not fully satisfactory since the response rate of some 35% is not superior to what is achieved with chemotherapy alone, however the 2-year survival rate is practically doubled at some 30%–35% but remains rather similar to what is observed with immunotherapy alone. Thus, other strategies are worth discussing. In the TPExtreme study, a first line chemotherapy with a taxane (TPEx) followed by second line with immunotherapy yields a 21.9 month median survival [37]. In the end, in order to determine which would be the best strategy, a randomized trial should be conducted comparing TPEx followed by immunotherapy vs. pembrolizumab + chemotherapy in first line. The reverse sequence, i.e., immunotherapy followed by chemotherapy would also be of interest since increased responses with chemotherapy have been reported after immunotherapy [38].

### 4.2. Studies Combining an Anti-PD-L1 with an Anti-CTLA-4

Immune escape pathways PD-L1/PD1 and CTLA-4 are non-redundant: the PD-L1/PD-1 pathway is mainly activated from peripheral tissues and inhibits T CD8+ cells while inducing differentiation of Treg cells, whereas the CTLA-4 pathway is activated from lymph nodes and inhibits T CD4+ cells [39]. The combination of an anti-PD-L1 with an anti-CTLA-4 is clinically more active than any of the two modes used alone in several types of cancers [40,41,42].

#### 4.2.1. Nivolumab + Ipilimumab vs. EXTREME: CHECKMATE-651 Study (NCT02741570)

This study randomized 950 patients, with a 1:1 ratio, between an experimental arm combining an anti-PD-1, nivolumab, with an anti-CTLA-4, Ipilimumab vs. a standard CT of the EXTREME type, primary endpoint being overall survival. Supposing however that this study comes out positive, it will not be possible to know if the same result could have been achieved with Nivolumab alone. Results are awaited however, there is already a negative signal delivered by study CHECKMATE-714 (NCT 02823574) comparing nivolumab + ipilimumab vs. nivolumab for platinum-sensitive and platinum-resistant patients with rate and duration of response as primary endpoint. Indeed, Bristol-Myers-Squibb put out a press-release informing that the primary endpoint had not been met without giving any further details [43]. Final results are also awaited.

#### 4.2.2. Durvalumab + Tremelimumab vs. Durvalumab vs. EXTREME: the KESTREL Study (NCT02551159)

This is a three arms study which will fully allow the evaluation of the impact of adding an anti-CTLA-4 (Tremelimumab) to an anti-PD-L1 (durvalumab) vs. EXTREME. Platinum-sensitive patients have been randomized with a 1:1:1 ratio (D + T, D and EXTREME). Results are awaited, but here again remain the negative signals concerning the efficacy of adding tremelimumab as delivered by CONDOR and EAGLE.

## 5. Future Developments

Thus immunotherapy has allowed significant progresses to be made, mainly by achieving durable responses with a favorable toxicity profile. Response rates however remain moderate most frequently below 20%. There are two avenues for possible improvement: discovery of new immunotherapy agents and of new combinations either of pure immunotherapies, or with targeted therapies. Only Phase Ib-II studies which have led to a randomized phase III are described here below (Table 5)**.**

### 5.1. New Immunotherapies and Combinations

#### 5.1.1. GSK609: T Cells Inducible Co-Stimulatory Receptor (ICOS) Agonist

GSK609 is a monoclonal anti-body IgG4 with an agonist activity on the ICOS receptor involved in the proliferation, differentiation and survival of T cells. ICOS is highly up-regulated upon T cells receptor stimulation [44] and is expressed on tumor infiltrating lymphocytes in many tumors [45]. Anti-tumor activity observed with an ICOS agonist is further enhanced with CTLA-4 and PD-1 blockade in non-clinical models [46,47].

An expansion phase Ib with GSK609 alone, then a phase II combining GSK609 with pembrolizumab were conducted in HNSCC and reported to ESMO 2019 [48] and later to ASCO 2020 [49]. Expansion phase Ib included 17 patients presenting an HNSCC pretreated in the R/M setting: 1 objective response (6%) and four stabilizations were observed for a disease control rate (DCR) of 31% showing a clinical activity of the GSK609 alone.

Phase II results of GSK609 combined with pembrolizumab, updated at ASCO 2020, show that for 34 patients presenting a HNSCC pretreated in 52% of the cases in the R/M setting but anti PD-1 naïve, 9 objective responses (26%) and 14 stabilizations were observed (DCR= 68%). Median PFS is 4.2 months and median survival is 13.1 months, which compares favorably with what is achieved with an anti-PD-1 for patients pretreated with platinum (Table 5). Moreover, the toxicity profile is very favorable with only 6% of treatment related Grade ≥ 3 AEs.

**Table 5 cancers-12-02691-t005:** Future developments with new IO agents and new combinations in R/M pretreated patients.

Study	IO Agents + Targeted Therapy	Pts Characteristics.	Nb of Pts	ORR (95%CI)	DCR (95%CI)	PFS Median (95% CI)	OS Median (95%CI)	Related Grade ≥ 3 Toxicity	Consecutive Phase III
INDUCE-1 NCT02723955 [49]	GSK 609 + pembro.	anti-PD-(L)1 naïve pretreated: 52%	34	24% (10.7–41.2)	68% (49.5–82.6)	4.2 mo (2.4–6.2)	13.1 mo (6.7–20)	6%	INDUCE-3: Pembro +/− GSK609 INDUCE-4: Pembro + CT+/− GSK609
NCT02643550 Expansion cohort 2 [50]	Monalizumab + Cetuximab	Platinum and Anti-PD-(L)1 pretreated	40	20% (10.5–34.8)	57.5%			42% 2% related to monalizumab	INTERLINK-1: cetuximab +/− monalizumab
NCT02501096 [51]	Pembro. + lenvatinib	HNSCC cohort Phase II ≤2 prior lines	22	46% (24.4–67.8)	90.9%	4.7 mo (4.0–9.8)		67%	LEAP-010 study NCT04199104

IO: Immuno Oncology; Nb of pts: Number of Patients; ORR: Overall Response Rate; DCR: Disease Control Rate; PFS: Progression Free Survival; OS: Overall Survival;. Pembro: pembrolizumab; CT: chemotherapy.

A randomized phase II study is in progress comparing in first line R/M for CPS ≥ 1 patients, a treatment with pembrolizumab + GSK609 vs. pembrolizumab + placebo (INDUCE-3 NCT04128696). A phase III is already planned which will compare the present standard pembrolizumab + platinum/5FU vs. pembrolizumab + platinum/5FU + GSK609 (INDUCE-4 NCT04428333).

#### 5.1.2. Monalizumab

Monalizumab is a first-in-class humanized IgG4 checkpoint inhibitor targeting the NKG2A receptor which is expressed on CD8+ T cells and NK cells. Blocking antibodies to NKG2A unleashed the reactivity of these effectors resulting in tumor control in multiple mouse models and early clinical trials [52].

##### Monalizumab alone and in combination with durvalumab: cohort I1 and I2 of the UPSTREAM trial (NCT03088059)

The UPSTREAM trial comprises 2 immunotherapy cohorts for platinum resistant patients in the R/M setting who are non-eligible for targeted therapy cohorts: cohort I1 Monalizumab alone and cohort I2 for PD-(L)1 pretreated patients: monalizumab + durvalumab vs. SoC. Results of cohort I1 have been reported to ESMO 2019 [53]: 25 patients were included 58% of whom had been pretreated with an anti-PD-(L)1, there were not any objective response and 6 patients were stabilized (23%) for a median duration of 3.8 months. Median PFS came out at 7.4 weeks and median survival at 6.7 weeks. Cohort I1 was discontinued for futility but inclusions are still being made in cohort I2, and the more so since there is a signal of activity for patients of cohort I1 pretreated with an anti-PD-(L)1 with a median survival of 8.6 months and a 1-year survival rate of 25%.

##### Monalizumab and Cetuximab

Cetuximab inhibits oncogenic EGFR signaling and binds to CD16/FcγRIII to promote ADCC. NK cell stimulation with monalizumab may enhance ADCC induced by cetuximab and thereby provide greater anti-tumor activity than cetuximab alone [54]. The combination of monalizumab and cetuximab has been studied in a multi-centric phase Ib-II trial (NCT02643550), the results of the phase II cohort concerning platinum resistant patients 45% of which had received an anti-PD-(L)1 are particularly interesting [55]: over 40 patients, a rate of 27.5% of objective response is achieved with a median PFS of 4.5 months and a median survival of 8.5 months. Results of an expansion cohort concerning patients who had all been treated with platinum and immunotherapy confirm a 20% ORR with a median duration of 5.2 months [50]. These results have led to planned a phase III (INTERLINK-1) for HNSCC patients in the R/M setting pretreated with platinum and anti-PD-(L)1 randomizing cetuximab + monalizumab vs. cetuximab + placebo.

### 5.2. Immunotherapy and Targeted Therapies

The most interesting targeted therapies which may be combined with immunotherapy, are probably the angiogenesis inhibitors because the vascular endothelial growth factor (VEGF) induces immune suppression. Indeed VEGF promotes the expansion of suppressive immune cells such as Treg or myeloid derived suppressor cells, suppresses effector T cells development, recruits tumor associated macrophages and inhibits the maturation and stimulatory functions of dendritic cells [56]. Among the angiogenesis inhibitors, lenvatinib, a multi-targeted tyrosine kinase inhibitor of VEGF receptor 1–3 among others, is one of the most effective. Studies in mouse tumor models showed that treatment combined with an anti-PD-1 monoclonal antibody demonstrated superior anti-tumor activity compared with either compound alone [57]. In a phase Ib-II study of a combination of lenvatinib with pembrolizumab, the HNSCC cohort ORRweek24 was 36% and the ORR was 46% with a median DOR of 8.2 months and a median PFS of 4.2 months [51].

Such results have justified the launching of a phase III trial randomizing the association pembrolizumab + lenvatinib vs. pembrolizumab + placebo with a planned recruitment of 500 patients presently underway in eight countries (LEAP-010 study NCT04199104) [58].

## 6. Conclusions

Immunotherapy, most particularly anti-PD-(L)1, has led to a significant breakthrough in the treatment of HNSCC in the R/M setting, an area where the prior major improvement dated back to 2008 with Cetuximab and the EXTREME regimen. Not only platinum-resistant patients may be treated efficiently with anti-PD-(L)1, but also for platinum-sensitive CPS ≥ 1 patients, pembrolizumab or the association pembrolizumab + platinum/5Fu have become the new first line standard. Although the median survival increase remains relatively moderate, the durability of responses allows an increase of the 2-year survival, which is almost doubled, be it for platinum resistant or platinum sensitive patients with a favorable toxicity profile added. Response rates remain however moderate being inferior to 20%, the association with chemotherapy allows an increase of the response rate which remains similar to what is achieved with chemotherapy alone and with survival rates barely superior to what is achieved by anti-PD-1 alone. A therapeutic sequences trial is still lacking in order to determine which is the best strategy: concomitant immuno-chemotherapy, immunotherapy followed by chemotherapy or the reverse sequence. Further improvements require imperatively an increase of the response rate. Two phase III studies are presently investigating the association of an anti-PD-(L)1 with an anti-CTLA-4 in first line, results are awaited, however other studies, conducted in different set-ups, have sent negative signals which might indicate that this combination may not work in HNSCC. Other phase III studies are either underway or planned to investigate new monoclonal antibodies such as GSK609 or monalizumab associated with anti-PD-(L)1 or cetuximab. Concerning the association with targeted therapies, the most promising probably is an association with angiogenesis inhibitors, presently lenvatinib which is being studied in LEAP-010 study. It is also necessary to make a better selection of patients in a position to benefit best from immunotherapy. The only predictive marker presently at our disposal is the expression of PD-L1 according to the CPS which when inferior to 1 concerns a group of about 15% of patient unsuited to anti-PD-1 therapy. The tumor mutation burden, as in lung cancers, seems to be promising.

## Figures and Tables

**Table 3 cancers-12-02691-t003:** KEYNOTE-048: Pembrolizumab vs. EXTREME according to CPS.

Nb of Pts		All pts (ITT)		CPS ≥ 1		CPS ≥ 20		CPS: 1–19		CPS < 1	
		P	EXTREME	P	EXTREME	P	EXTREME	P	EXTREME	P	EXTREME
		301	300	257	255	133	122	124	133	44	45
OS	med.	11.5	10.7	12.3	10.3	14.8	10.7	10.8	10.1	7.9	11.3
	HR (95% CI)	0.83 (0.70–0.99) *p* = 0.019		0.74 (0.61–0.90) *p* = 0.00133		0.58 (0.44–0.78) *p* = 0.00010		0.86 (0.66–1.12) *p* = 0.1282		1.51(0.96–2.37) *p* = 0.96	
	2-year	27%	18.8%	28.9%	17.4%	35.3%	19.1%	20.5%	19.0%	15.9%	26.7%
PFS	med.	2.3	5.2	3.2	5.0	3.4	5.3	2.2	4.9	2.1	6.2
	HR (95% CI)	1.29 (1.09–1.53) *p* = 0.998		1.13 (0.94–1.36) *p* = 0.895		0.99 (0.76–1.29) *p* = 0.4679		1.25 (0.96–1.61) *p* = 0.95		4.31 (2.63–7.08) *p* = 1.00	
	1-year	17.6%	15%	20.6%	13.6%	23.5%	15.1%	24.2% ^§^	41.4% ^§^	11.4% ^§^	56% ^§^
RESPONSE	ORR	16.9%	36.0%	19.1%	34.9%	23.3%	36.1%	14.5%	33.8%	4.5%	42.2%
	SD	27.2%	34.0%	28.0%	32.9%	30.1%	35.2%				
	PD	40.5%	12.3%	38.9%	12.9%	31.6%	9.8%				
	DOR	22.6 (1.5–43)	4.5 (1.2–38.7+)	23.4 (1.5–43+)	4.5 (1.2–38.7+)	22.6 (2.7–43+)	4.2 (1.2–31.5)	NR (1.5+–38.9+)	5.0 (1.4+–38.7+)	2.6 (2.2–3.0)	7.8 (2.0–38.6+)
	≥6 mo.	77.8%	38.8%	81.1%	36%	83.5%	34.8%	76.5%	36.6%	0%	52.7%

CPS: Combined Positive Score; pts: patients; ITT: Intent To Treat; P: Pembrolizumab; Nb: Number; OS: overall survival; med.: median in months; HR: Hazard Ratio; PFS: Progression Free Survival; ^§^: 6-month PFS; ORR: Overall Response Rate; SD: Stable Disease; PD: Progressive Disease; DOR: Duration Of Response in months (extremes); ≥6 mo: % DOR≥6 mo.

**Table 4 cancers-12-02691-t004:** KEYNOTE-048: Pembrolizumab + Chemotherapy (C) vs. EXTREME according to CPS.

Nb of Pts		All pts (ITT)		CPS ≥ 1		CPS ≥ 20		CPS: 1–19		CPS < 1	
		P + C	EXTREME	P + C	EXTREME	P + C	EXTREME	P + C	EXTREME	P + C	EXTREME
		281	278	242	235	126	110	116	125	39	43
OS	med.	13.0	10.7	13.6	10.4	14.7	11.0	12.7	9.9	11.3	10.7
	HR (95% CI)	0.72 (0.60–0.87) *p* = 0.00025		0.65 (0.53–0.80) *p* = 0.00002		0.60 (0.45–0.82) *p* = 0.00044		0.71 (0.54–0.94) *p* = 0.00726		1.21 (0.76–1.94) *p* = 0.78	
	2-year	29.4%	18.2%	30.8%	16.8%	35.4%	19.4%	25.9%	14.5%	20.5%	25.6%
PFS	med.	2.3	5.2	5.1	5.0	5.8	5.3	4.9	4.9	4.7	6.2
	HR (95% CI)	1.29 (1.09–1.53) *p* = 0.998		0.84 (0.69–1.02) *p* = 0.0369		0.76 (0.58–1.01) *p* = 0.0295		0.93 (0.71–1.21) *p* = 0.29		1.46 (0.93–2.30) *p* = 0.94	
	1-year	17.6%	15%	19.7%	12.5%	23.9%	14.0%	40.1% ^§^	40.0% ^§^	43.6% ^§^	53.8% ^§^
RESPONSE	ORR	35.6%	36.3%	36.4%	35.7%	42.9%	38.2%	29.3%	33.6%	30.8%	39.5%
	SD	27.8%	34.2%	26.4%	32.8%	23.0%	34.5%				
	PD	17.1%	11.9%	17.4%	12.3%	15.1%	8.2%				
	DOR	6.7 (1.67–39.07)	4.3 (1.2+–31.5+)	6.7 (1.6+–39.0+)	4.5 (1.2–38.7+)	7.1 (2.1+–39.0+)	4.2 (1.2+–31.5+)	5.6 (1.6+–25.6+)	5.(1.4+–38.7+)	5.7 (2.6–20.6+)	4.3 (2.0–31.2+)
	≥6 mo.	53.5%	36.8%	54.3%	34.3%	60.2%	34.0%	44.3%	34.0%	46.9%	49.0%

CPS: Combined Positive Score; pts: patients; ITT: Intent To Treat; P: Pembrolizumab; Nb: Number; OS: overall survival; med.: median in months; HR: Hazard Ratio; PFS: Progression Free Survival; ^§^: 6-month PFS; ORR: Overall Response Rate; SD: Stable Disease; PD: Progressive Disease; DOR: Duration Of Response in months (extremes); ≥6 mo: % DOR≥6 mo.

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
