# Peer review of "Immunotherapy Breakthroughs in the Treatment of Recurrent or Metastatic Head and Neck Squamous Cell Carcinoma"

_cancers, 2020, doi:10.3390/cancers12092691_

Round 1
Reviewer 1 Report
The Authors performed a comprehensive and clear review about immunotherapy in head and neck squamous cell carcinoma. They should be congratulated.
Just a few comments and suggestions:
- platinum-resistant scenario. I would also cite the studies with real-world data presented till now (Hanna, Topnivo, etc), to give more data outside clinical trials
- combination of anti PD-1 or PDL-1 and anti CTLA-4: not satisfactory results till now should be discussed and mechanisms that do not allow synergistic effect could be speculated
- the Authors should leave open the door about a sequencing strategy, with chemotherapy followed by immunotherapy (I suggest discussing the possible role of taxane in priming better immunotherapy effect, see results with TPEx presented at ASCO 2020) or with immunotherapy followed by chemo (see in this regard a few papers published regarding increased chemo effect after immune priming)
- the role of tumor mutational burden as possible predictor of immunotherapy response should be discussed a little bit more, also showing the possible criticism in this regard
- the brief paragraph about immunotherapy in locally advanced setting is out of the scope of this review and it is so short that cannot be even discussed. I suggest taking off this paragraph (and the conclusions referred to this), as the paucity of data and discussion worsens the high quality of the review
Author Response
Thank you for your comments:
Comment 1: For platinum resistant patients treated outside clinical trials, I shall cite the TOPNIVO study (C. Even ASCO 2019 abstr 6032)
Comment 2: I shall add several suppositions in order to explain why anti-CTLA-4 combined to anti-PD-L1 does not seem to be synergistic in spite of a solid scientific rational. lymph nodes surgical resection or irradiation might impair the anti-CTLA-4 action. Furthermore, Tremelimumab being an IgG2 is not able to induce anti-body-dependant cell toxicity whereas NK cell are the most represented T-cells in the TME of HNSCC (M. Merlano, Ann Oncol 2020).
Comment 3: I shall add the following paragraph in my manuscript: PFS2 (cumulative progression free survival of the two first lines) as observed in the KEYNOTE-048 study, seems to demonstrate that a second line treatment particularly with immunotherapy following a first line chemotherapy may not be as efficient as an immunotherapy combined with a chemotherapy as early as first line. However, KEYNOTE-048 is not a therapeutic sequences study, less of 50% of patients have received a second line and 16% only received an immunotherapy. In the TPExtreme study (J. Guigay ASCO 2020), a first line chemotherapy with a taxane (TPEx) followed by second line with immunotherapy yields a 22.4 month median survival. In the end, in order to determine which would be the best strategy, a randomized trial should be conducted comparing TPEx followed by immunotherapy vs Pembrolizumab + Chemotherapy in first line. The reverse sequence, ie immunotherapy followed by chemotherapy, would also be interesting since increased responses to chemotherapy have been reported after immunotherapy (K. Saleh, Eur. J. Cancer, 121 (2019) 123-129).
Comment 4: I shall add the following paragraph in my manuscript:The bTMB post-hoc analysis (EAGLE study) confirms the findings of a retrospective study in 126 HNSCC patients where the response rate and overall survival were correlated with TMB values ( TMB: 21.3 mut/MB in any responders vs 8.2 mut/MB in non responders p=0.01 and median survival: 20 months if TMB >10 mut/MB vs 6 months if TMB<5 mut/MB p=0.01 (Hanna GJ, JCI insight 2018 3(4); e98811). In addition, TMB and inflammatory biomarkers are independent predictors of response and survival with pembrolizumab ( Critescu R, Science 2018, 362 (6411). eaar 3593). However the TMB cut-off has to be more precisely defined and a prospective trial has to be conducted in order to validate his predictive value.
Comment 5: The brief paragraph about immunotherapy in locally advanced setting and the conclusion referred to this will be deleted.
Reviewer 2 Report
This paper is a complete overview on what's going on actually with the use of ICI and immunotherapy for RM/HNSCC. This paper is really well writen and give all the informations that clinicians need for their daily practice and to understand changes due to immuntherapy use in the field of RM/HNSCC. This is a really nice paper and authors should be congratulated for this.
Comment 1 : for Keynote 040 study, lines 155-156 : could the authors give some détails on patients cisplatine sensitive and cisplatine refractory included in the study ?
Comment 2 : Keynote 048 lines 267 to 271 : could the authors comment on the excess of PD with Pembrolizumab alone, could it be in relation with the possibility of hyperprogression like it has been reported in several papers (example : Saada-Bouzid E in Annals of oncology 2017)
Comment 3 : Keynote 048 : lines 290 to 294 and 316 to 318 : do the authors think that PFS2 results are really relevant and add some information to the intial results ?
Comment 4 : in the conclusion could the authors give some perspective of other first line treatments for patients with RM/HNSCC especially for tumors with CPS < 1 ot between 1 and 19, is there an opportunity for chemotherapy alone ? without immunotherapy ? is there a role for new chemo combination like Cisplatinum + taxane + Cetuximab (J. Guigay ASCO 2019).
Author Response
Thank you for comments
Comment 1: in the KN-040 study, 14% to 16% of enrolled patients showed disease progression within 3-6 months of a previous cisplatin based multimodal therapy for locally advanced disease, 57% of patients had received a first line platinum based chemotherapy in the R/M setting some of whom were considered as platinum sensitive if they had progressed more than six months after the last dose of platinum, and finally 27% to 28% are in second or third line in the R/M setting. I shall precise this point in my manuscript.
Comment 2: The high rate of disease progression (40.5%) in the pembrolizumab alone arm of the KEYNOTE-048 could be partly explained by the phenomenon of hyperprogression where the disease seems to grow faster after initiation of immunotherapy. Hyperprogression has been described in several tumour types, rates of hyperprogression range from 4% to 29% depending on the definition and the methods used to assess tumour growth kinetics (E. Borcoman, Ann. Oncol. 2019, 30, 385-396). In HNSCC it may be as high as 29.5% (E. Saada-Bouzid, Ann. Oncol. 2017) with the consequence of having to change treatment quickly in the face of rapid tumour progression, especially in case of clinical deterioration. However the concept of hyperprogression is controversial, given that it is not possible to determine whether the acceleration of growth kinetics is due to immunotherapy or reflect the natural history of cancer. I shall precise this point in my manuscript
Comment 3: PFS2 (cumulative progression free survival of the two first line) as observed in the KEYNOTE-048 study, seems to demonstrate that a second line treatment particularly with immunotherapy following a first line chemotherapy may not be as efficient as an immunotherapy combined with a chemotherapy as early as first line. However, KEYNOTE-048 is not a therapeutic sequences study, less of 50% of patients have received a second line and 16% only received an immunotherapy. In the TPExtreme study (J. Guigay ASCO 2020), a first line chemotherapy with a taxane (TPEx) followed by second line with immunotherapy yields a 22.4 month median survival. In the end, in order to determine which would be the best strategy, a randomized trial should be conducted comparing TPEx followed by immunotherapy vs Pembrolizumab + Chemotherapy in first line. I shall precise this point in my manuscript
Comment 4:
- For CPS <1, immunotherapy is non indicated either with chemo or not. EXTREME or better TPEx should be preferred.
- For CPS 1-19, the EMA report shows a significant benefit in overall survival in the pembrolizumab + chemotherapy arm vs EXTREME, so pembrolizumab + chemo should be preferred. Pembrolizumab alone is not superior to EXTREME but for low risk patients it could be an interesting option because of long lasting responses and low toxicity.
- Regarding the role of TPEx see response to comment 3